# Influence of the Surface Chemistry of Graphene Oxide on the Structure–Property Relationship of Waterborne Poly(urethane urea) Adhesive

**DOI:** 10.3390/ma14164377

**Published:** 2021-08-05

**Authors:** Abir Tounici, José Miguel Martín-Martínez

**Affiliations:** Adhesion and Adhesives Laboratory, Department of Inorganic Chemistry, University of Alicante, 03080 Alicante, Spain; abirbibo27@gmail.com

**Keywords:** waterborne poly(urethane urea), graphene oxide derivative, surface chemistry, adhesion, micro-phase separation, structure–property relationship

## Abstract

Small amounts—0.04 wt.%—graphene oxide derivatives with different surface chemistry (graphene oxide—GO-, amine-functionalized GO—A-GO-, reduced GO—r-GO) were added during prepolymer formation in the synthesis of waterborne poly(urethane urea) dispersions (PUDs). Covalent interactions between the surface groups on the graphene oxide derivatives and the end NCO groups of the prepolymer were created, these interactions differently altered the degree of micro-phase separation of the PUDs and their structure–properties relationships. The amine functional groups on the A-GO surface reacted preferentially with the prepolymer, producing new urea hard domains and higher percentage of soft segments than in the PUD without GO derivative. All GO derivatives were well dispersed into the PU matrix. The PUD without GO derivative showed the most noticeable shear thinning and the addition of the GO derivative reduced the extent of shear thinning differently depending on its functional chemistry. The free urethane groups were dominant in all PUs and the addition of the GO derivative increased the percentage of the associated by hydrogen bond urethane groups. As a consequence, the addition of GO derivative caused a lower degree of micro-phase separation. All PUs containing GO derivatives exhibited an additional thermal decomposition at 190–206 °C which was ascribed to the GO derivative-poly(urethane urea) interactions, the lowest temperature corresponded to PU+A-GO. The PUs exhibited two structural relaxations, their temperatures decreased by adding the GO derivative, and the values of the maximum of tan delta in PU+r-GO and PU+A-GO were significantly higher than in the rest. The addition of the GO derivative increased the elongation-at-break, imparted some toughening, and increased the adhesion of the PUD. The highest T-peel strength values corresponded to the joints made with PUD+GO and PUD+r-GO, and a rupture of the substrate was obtained.

## 1. Introduction

Waterborne polyurethane adhesives are used in food packaging, paper sizing, textile, wood, automobile, footwear, and biomedical industries [1] due to their excellent elasticity, high adhesion, and resistance to low temperatures [2]. The adhesive properties of the waterborne polyurethanes are determined by the segmented structure of hard and soft domains of the ionomeric poly(urethane urea) dispersion (PUD). The soft domains are made of polyol chains and impart flexibility, while the hard domains are made by reacting the diisocyanate and the chain extender, and they provide the mechanical properties [3]. The incompatibility of the soft and hard domains results in micro-phase separation; hydrogen bonding between the hard and soft domains is generally produced. The degree of micro-phase separation of the PUD depends on the nature of the reactants and the synthesis procedure, among other parameters.

Graphene oxide (GO) is generally produced by chemical oxidation of graphite [4]. It has a high specific surface area, a high modulus, and a high thermal conductivity [5]. The surface chemistry on the basal plane and edges of GO consist of phenolic hydroxyl (-OH), epoxy (C-O-C), carboxyl (-COOH) and carbonyl (-C=O) groups [6]. These groups may covalently react with several polymers to produce nanocomposites. The amount of oxygen functional groups on the GO nano-sheets can be partially removed by treatment with reduction agents such as hydrazine, the reduced GO (r-GO) is less hydrophilic than GO. On the other hand, nitrogen functional groups can be grafted on the GO surface by treatment with amines, and the higher nucleophile properties of nitrogen with respect to oxygen may facilitate the interaction with several polymers [7].

The presence of functional groups in graphene-based materials is important for medical application, especially those related to drug/gene/protein delivery systems and materials with antimicrobial properties [8]. N-doped graphene has a potential in electrochemical applications, particularly the development of anode or cathode materials for supercapacitors, metal–air batteries, and fuel cells [9]. On the other hand, the GO based nano filtration membranes enhance the dye rejection and antifouling property in wastewater treatment and seawater desalination [10].

Several GO derivatives have been added to PUDs for preparing nanocomposites with improved mechanical and thermal properties. Yoon et al. [11] have synthesized hybrid acrylic-PUD/allyl isocyanate modified GO nanocomposites by UV curing, the composites showed improved tensile strength and glassy and rubbery moduli, and high thermal stability. Zheng et al. [12] have synthesized PUDs containing 0.50 wt.% isocyanate functionalized GO via in-situ polymerization with improved thermal stability, tensile strength, and hydrophobicity. These improved properties were ascribed to the covalent bonds of the isocyanate groups on the GO nano-sheets and the polyurethane chains. On the other hand, Chen et al. [13] synthesized PUDs with 0.5 and 1 wt.% r-GO, TiO_2_/r-GO and surfactant-modified TiO_2_/r-GO, the highest thermal stability and elongation-at-break were obtained by adding r-GO. Furthermore, Wang et al. [14] have shown that the addition of 2 wt.% 3-aminopropyl triethoxysiloxane functionalized graphene increased the tensile strength and Young modulus of PUDs. Wu et al. [15] added 0.1–1 wt.% amine-functionalized reduced GO to PUD in order to improve its mechanical and thermal properties, the thermal conductivity also increased along with the increase of the functionalized GO amount. Similarly, Hu et al. [16] added amine functionalized GO to improve the thermal, mechanical, and hydrophobic properties of PUDs, and the enhanced properties were ascribed to the creation of interfacial interactions. Zhang et al. [17] have also shown that the addition of triethylene tetramine + polyethylene glycol diglycidyl ether functionalized GO enhanced the water resistance, the thermal stability, and the mechanical properties of PUDs. Furthermore, different GO derivatives have been used to improve the anticorrosive properties of coatings made from PUDs [18,19].

The addition of GO derivatives to improve the adhesion properties of waterborne polyurethane adhesives and coatings has also been studied. Zhao et al. [20] prepared 1 wt.% polydopamine functionalized graphene/PUD coatings to increase the adhesive strength and the water contact angle values. The adhesion, mechanical, and thermal properties of the PUD coatings modified with cellulose nanocrystals and graphene have been studied by Yang et al. [21]. Kale et al. [22] added epoxy functionalized GO and amine functionalized nanosilica to PUD by physical mixing, resulting in the improvement of the mechanical, thermal, and adhesion to leather properties. Cristofolini et al. [23] added 0.01 wt.% carboxyl-functionalized graphene nano-platelets (GP) + 0.1–1 wt.% GO to commercial PUD by vigorous mixing for increasing the adhesion properties, the addition of 0.01 wt.% GP increased the adhesive strength. Tounici and Martín-Martínez [24,25] added 0.01 to 0.10 wt.% GO during the synthesis of PUDs, and improved adhesive strength was obtained by adding 0.04 wt.% GO. The improved adhesion was ascribed to the changes in the degree of micro-phase separation between the hard and soft domains induced by the covalently bonded oxygen functional groups on the GO sheets and the isocyanate groups during the synthesis of the PUDs.

Although the mechanical and thermal properties of PUDs containing different graphene derivatives have been previously studied, the influence of the surface chemistry of the GO on the structure, viscoelastic, and adhesion properties of the PUDs have been scarcely addressed. Most previous studies have dealt with PUDs containing more than 0.10 wt.% graphene derivative, even the excellent performance of the PUDs containing less than 0.10 wt.% GO have been recently demonstrated [24,25], the optimal performance was obtained by adding 0.04 wt.% GO before prepolymer formation during PUD synthesis. Therefore, in this study, the surface chemistry of GO was changed by decreasing the number of functional groups (reduced GO) and by grafting nitrogen functionality (amine-functionalized GO) on the GO nano-sheets. Furthermore, 0.04 wt.% each GO derivative was added during the synthesis of the PUDs and the changes on the structure–property relationship were assessed paying particular attention to the adhesive properties.

## 2. Materials and Methods

### 2.1. Materials

Different reactants were used for synthesizing the waterborne poly(urethane urea) dispersions (PUDs). Dried polyadipate of 1,4-butanediol with molecular weight of 2000 Da—Hoopol F-501 (Synthesia, Barcelona, Spain)—was used as polyol. Isophorone diisocyanate (IPDI, 98 wt.% purity) was used. As internal emulsifier, 2,2 bis(hydroxymethyl) propionic acid (DMPA, 98 wt.% purity) was used, triethylamine (TEA, 99 wt.% purity) was used as neutralization agent, monohydrated hydrazine (HZ, 60 wt.% purity) was used as chain extender, and dibutyltindilaurate (DBTDL, 95 wt.% purity) was used as catalyst—all were supplied by Sigma Aldrich (Barcelona, Spain). Acetone (99.5 wt.% purity)—Sigma Aldrich, St. Louis, MO, USA—and de-ionized water were used.

Three different graphene oxide derivatives were used: Graphene oxide (GO), reduced graphene oxide (r-GO), and amine-functionalized graphene oxide (A-GO)—all were supplied by Graphenea (San Sebastián, Spain). A-GO was obtained by treatment of GO with dodecyl amine. Before use, all GO derivatives were dried in oven at 80 °C overnight and they were kept in oven at 80 °C until use.

### 2.2. Synthesis of the PUDs

Different PUDs without and with 0.04 wt.% GO derivative were synthesized by using the acetone method. This amount of the GO derivatives was selected as optimal in previous study [24]. A total 5 wt.% DMPA (with respect to the total amount of prepolymer) was added, the targeted solids content was 40 wt.%, and NCO/OH ratio of 1.5 was used (both OH groups of the polyol and DMPA were considered). The PUD without GO derivative (reference) was synthesized by following the different stages given in Figure 1.

The polyol, DMPA, and the catalyst were added into the reactor at 80 °C and stirred at 450 rpm for 30 min. Then, IPDI diisocyanate was added slowly at 80 °C under stirring and the reaction lasts for 3 h to obtain a prepolymer. The temperature was lowered to 42 °C and acetone was added to dissolve the prepolymer and reduce its viscosity; 30 min later, TEA in 25 mL acetone was added to neutralize the protons of the DMPA in the prepolymer by stirring at 40 °C and 450 rpm for another 30 min. The chain extension was carried out with hydrazine at 40 °C under stirring at 450 rpm for 30 min. The stirring speed was increased to 900 rpm, and water was added maintaining the stirring at 40 °C for 30 min. Finally, the residual acetone was removed in rotavapor (Büchi B-210, Flawil, Switzerland) at 50 °C under 300 mbar for 1 h.

The PUDs with 0.04 wt.% GO derivative (GO, A-GO, r-GO) were synthesized similarly to the reference PUD, but the GO derivative was added to the polyol before being added in the reactor for prepolymer formation. A mixture of 20 g polyol and 0.04 g of each GO derivative were placed inside a closed polypropylene container which in turn was placed in a double orbital Speed Mixer centrifuge (Hauschild Engineering, Hamm, Germany) operating at 2010 rpm for 180 s. Then, the polyol+GO derivative mixture was heated on a heating plate at 70 °C for 30 min to remove the residual water.

Figure 2 shows the appearance of the PUDs after one month of their syntheses. The color of PUD+A-GO and PUD+r-GO are darker than that of PUD+GO.

Several properties were measured in solid poly(urethane urea) (PU) films, obtained by placing 12 g PUD in square silicone mold of dimensions 10 cm × 10 cm. The water was removed at room temperature for four days followed by heating at 40 °C for 8 h. The thicknesses of the PU films were about 1.4 mm.

Thinner PU films for stress-strain tests were prepared by placing 10 g PUD on glass plate of dimensions 12 mm × 24 mm coated with Teflon^®^ sheet. Three pieces of double side tape (3M, St. Paul, MN, USA) were placed over the sides of the mold in order to adjust a thickness of 200 µm. Once the PUD was spread over the mold, the water was left to evaporate for four days at room temperature followed by heating at 40 °C for 8 h. The thicknesses of the PU films were about 60 µm.

### 2.3. Experimental Techniques

#### 2.3.1. Characterization of the Graphene Oxide (GO) Derivatives

X-ray photoelectron spectroscopy (XPS). The chemical compositions of the GO derivative surfaces were determined by XPS in a Surface Science SSX-100 ESCA spectrometer (Thermo Ficher Scientific, Waltham, MA, USA) by using an Al-Kα X-ray source. High-resolution C1s, O1s, and N1s spectra were collected and the binding energies were corrected at 285.0 eV for the main hydrocarbon peak. Two replicates for each graphene oxide derivative were carried out.

Thermal gravimetric analysis (TGA). The amounts of functional groups of the GO derivatives were determined in a TGA Q500 equipment (TA Instruments, New Castle, DE, USA). 10–15 mg of sample were placed in platinum crucible and heated under nitrogen (flow rate: 100 mL/min) from room temperature up to 800 °C, by using a heating rate of 10 °C/min.

X-ray diffraction (XRD). The crystallinities of the GO derivatives were determined in a Bruker D8-Advance diffractometer (Bruker, Ettlingen, Germany). The Kα wavelength (1.540598 A) of copper was used.

Raman spectroscopy. The Raman spectra of the GO derivatives were obtained in a Jasco NRS-5100 Raman spectrometer (Jasco, Madrid, Spain) using an excitation wavelength of 532 nm.

Transmission electron microscopy (TEM). The morphologies of the GO derivatives’ surfaces were determined in a Jeol JEM-1400 Plus instrument (Jeol, Tokyo, Japan) by using an acceleration voltage of 120 kV.

#### 2.3.2. Characterization of the Poly(urethane urea) Dispersions (PUDs) without and with GO Derivative

Solids content. The solid contents of the PUDs were determined in a DBS 60-3 thermo balance (Kern & Sohn GmbH, Balingen, Germany) by heating at 105 °C for 15 min followed by heating at 120 °C until constant mass was obtained.

pH measurement. The pH values of the PUDs were measured at 25 °C in a pH-meter PC-501 (XS Instruments, Carpi, Italy) equipped with XC-PC510 electrode.

Viscosity. The viscosities at 25 °C of the PUDs as a function of the shear rate were measured in a DHR-2 rheometer (TA Instruments, New Castle, DE, USA) using coaxial cylindrical geometry according to DIN 53019 standard. The gap was set to 2 mm.

#### 2.3.3. Characterization of the Poly(urethane urea)s (PUs) without and with GO Derivative

Attenuated total reflectance Fourier transform infrared (ATR-IR) spectroscopy. The ATR-IR spectra of the PUs were obtained in a Tensor 27 FT-IR spectrometer (Bruker Optik GmbH, Ettlinger, Germany) by using Golden Gate single reflection diamond ATR accessory.

Raman spectroscopy. The Raman spectra of the PUs were recorded in a Jasco NRS-5100 Raman spectrometer (Jasco, Madrid, Spain) by using an excitation wavelength of 632 nm (HeNe source).

Differential scanning calorimetry (DSC). The DSC traces of the PUs were carried out in a TA DSC Q100 V6.2 instrument (TA Instruments, New Castle, DE, USA) under nitrogen atmosphere. Closed aluminum pans containing 10–15 mg sample were heated from −70 to 110 °C by using a heating rate of 10 °C/min, followed by a cooling run to −80 °C and a second DSC heating run from −80 to 200 °C (heating rate = 10 °C/min). From the second DSC heating run, the glass transition temperatures (T_g_s) of the PUs were obtained.

X-ray diffraction (XRD). The crystallinities of the PUs were determined by wide angle X-ray diffraction in a Bruker D8-Advance equipment (Bruker, Ettlingen, Germany) provided with X-ray generator Kristalloflex K 760-80F (3000 W, 20–60 kV, current of 5–80 mA). A voltage of 40 kV and the Kα wavelength (1.5418 Å) of copper were used.

Thermal gravimetric analysis (TGA). The structure and the thermal properties of the PUs were assessed in a TGA Q500 equipment (TA Instruments, New Castle, DE, USA). 10–15 mg of PU were placed in platinum crucible and heated under nitrogen atmosphere from room temperature up to 800 °C, by using a heating rate of 10 °C/min.

Dynamic mechanical thermal analysis (DMA). The viscoelastic properties of the PUs were measured in a DMA-Q800 equipment (TA Instruments, New Castle, DE, USA) by using the single cantilever mode. PU films with dimensions 18 mm × 13 mm × 1 mm were used and they were heated from −100 to 70 °C under nitrogen atmosphere by using a heating rate of 5 °C/min. All experiments were carried out at a frequency of 1 Hz, an amplitude of 20 μm and a strain of 0.5%.

Stress-strain tests. The mechanical properties of the PUs were assessed by stress-strain tests according to ISO 37 standard. Dog-bone test specimens were cut, and the stress-strain tests were carried out in an Instron 4411 universal testing machine (Instron, Buckinghamshire, UK) provided with mechanical extensometer, a pulling rate of 100 mm/min was used. Five replicates were measured and averaged.

Water contact angle measurements. The contact angle measurements were measured at 21 °C in an ILMS goniometer (GBX Instruments, Bourg de Pèage, France), bi-distilled and deionized water was used. Water droplets of 4 µL were placed on different locations of the PU surface, and the contact angle values were averaged.

Confocal laser microscopy. A Leica TCS SP2 microscope (Leica, Heidelberg, Germany) was used to analyze the dispersion of the GO derivative nano-sheets in the PU matrices.

Scanning electron microscopy (SEM). PU films were immersed in liquid nitrogen for 2 min and mechanically fractured immediately after being taken out. The fractured surfaces of the PUs were gold coated and then analyzed in a Jeol JSM-840 SEM microscope (Jeol, Tokyo, Japan). The energy of the electron beam was 10 kV.

#### 2.3.4. Adhesion Properties

T-peel tests of plasticized poly (vinyl chloride) (PVC)/PUD adhesive/plasticized PVC joints. The adhesive strengths of the joints made with PUDs were obtained from T-peel tests of solvent-wiped plasticized PVC/PUD adhesive/solvent-wiped plasticized PVC joints (Figure 3). Plasticized PVC test samples with dimensions of 30 mm × 150 mm × 5 mm were wiped with ethyl ketone for plasticizer removal. 0.9 g PUD was applied by brush to each PVC strip, allowing water evaporation at 25 °C for 1 h followed by sudden heating at 80 °C for 10 s under infrared radiation (reactivation process). The PVC strips were immediately placed in contact and a pressure of 0.8 MPa was applied for 10 s to achieve a suitable joint [25]. The T-peel strengths were measured 1 and 72 h after joints formation in an Instron 4411 universal testing machine (Instron Ltd., Buckinghamshire, UK), a crosshead speed of 100 mm/min was used. Five replicates were measured and averaged. The loci of failure of the joints were assessed by visual inspection.

## 3. Results and Discussion

### 3.1. Characterization of the GO Derivatives

The amounts of the functional groups on the graphene oxide derivative surfaces (GO, A-GO, r-GO) obtained by XPS are shown in Table 1. The GO surface shows the lowest amount of carbon and the highest amount of oxygen, indicating higher degree of surface oxidation, and both r-GO and A-GO surfaces have similar amounts of carbon and oxygen, but the amount of nitrogen is somewhat higher in A-GO.

Appendix A shows the curve fittings of the high resolution C1s spectra of the GO derivatives; the ones of O1s and N1s of A-GO are shown, as representative examples, in Appendix A. The percentages of the carbon species on the GO, A-GO, and r-GO surfaces were assessed from the C1s spectra and they are shown in Table 2; the assignment of the carbon species was carried out according to references [26,27]. The GO surface contains the lowest percentage of C-C and the highest number of C-O groups. The percentages of C-O groups are similar in the r-GO and A-GO surfaces, and the r-GO surface shows the highest percentage of C=O groups. Therefore, all GO derivative surfaces show the same kind of surface carbon-oxygen functional groups but in different amounts, and, additionally, the A-GO surface contains 88 at.% C-N (sp^3^)—binding energy = 399.5 eV) and 12 at.% -NH_2_ (sp^2^) species—binding energy = 401.1 eV)—Appendix A.

The amounts of functional groups of GO, A-GO and r-GO were assessed by TGA. Figure 4 shows the variation of the weight as a function of the temperature of the GO derivatives. The amounts of functional groups are high and somewhat similar in GO (61 wt.%) and A-GO (56 wt.%), but r-GO shows only 18 wt.% functional groups. The nature of the functional groups in the GO derivatives was assessed from the derivative thermal gravimetric analysis plots (Appendix A). GO shows the thermal decompositions due to absorbed moisture (49 °C), -C-OH (158, 231–257 °C), and –O-C=O (676–688 °C) groups—Table 3, whereas A-GO shows the thermal decompositions of absorbed moisture (50 °C), -C-N (177 °C), -C-OH (263 °C), and C=O (361, 438 °C) groups. On the other hand, r-GO shows two thermal decompositions due to absorbed moisture (54 °C) and -C-OH groups (258 °C). The thermal decomposition of the C-OH groups appears at about 257–258 °C in all graphene oxide derivatives, but in GO some additional thermal decompositions at 158 and 231 °C of the C-OH groups also appear, indicating different interactions with the graphene sheets. Therefore, GO shows the highest percentage of -C-OH groups and A-GO shows the highest percentage of C=O groups; furthermore, A-GO contains 18 wt.% C-N groups due to the treatment of GO with dodecyl amine, in agreement with the findings of Wang et al. [28].

The X-ray diffractograms of the graphene oxide derivatives (GO, A-GO, r-GO) are shown in Figure 5. GO shows three main diffraction peaks at 2Ɵ values of 9.55, 18.30, and 25.35°, the one at 9.55° is characteristic of GO [29]. The peak at 2Ɵ value of 9.55° is associated to the (101) diffraction plane due to the formation of oxygen-containing groups in GO which causes an expansion of the C–C interplanar spacing of the graphite, and the two peaks at 2Ɵ values of 18.30 and 25.35° are associated to the oxidation of graphite [29,30]. On the other hand, the X-ray diffractogram of r-GO shows the varnishing of the peak at 2Ɵ value of 9.55° caused by the dismantle of the regular stacking of the GO sheets and the aggregation of the graphene sheets, the broad weak peak at 2Ɵ value of 20.75° is retained, indicating parallel stacking of the r-GO sheets [29]. Furthermore, the reduction of the GO removes the most oxygen containing groups in the interstitial graphitic spaces and restores the C–C lattice of the graphite and, therefore, the diffractogram of r-GO shows some peaks of graphite at 2Ɵ values of 42.9°—(101) plane -, 47.55°—(102) plane -, 56.55°—(004) plane -, and 77.1°—(110) plane [31]. The amine-functionalization of the GO also shows the varnishing of the peak at 2Ɵ value of 9.55° caused by the dismantle of the regular stacking of the GO sheets and the broad weak peak at 2Ɵ value of 19.15° is retained, indicating parallel stacking of the A-GO sheets; in addition, some aggregation of the graphene sheets in A-GO is evidenced by the diffraction peak at 2Ɵ value of 42.9° due to the (101) plane of the graphite [30].

The structural changes produced during the reduction and amine-functionalization of GO are evidenced in the Raman spectra (Figure 6). All Raman spectra show the D band at 1342 cm^−1^ related to the disordered structure of GO, and the G band at 1598 cm^−1^ related to C-C bond of sp^2^ carbon. The ratio of the intensities of the D and G bands (I_D_/I_G_) is related to the order of graphene nano-sheets, i.e., the increase of the I_D_/I_G_ ratio indicates a decrease of the average size of the C-C sp^2^ nano-sheets. The I_D_/I_G_ ratios in the Raman spectra of r-GO and A-GO are higher than in GO due to the presence of unrepaired defects after the removal of oxygen-containing functional groups. Thus, GO has the highest average size of the C-C sp^2^ nano-sheets whereas r-GO has the lowest. It is worthy to note that these I_D_/I_G_ ratios are consistent with the values in the literature [32,33].

The increase in the ordered structure of the graphene nano-sheets in A-GO and r-GO is also evidenced in the TEM micrographs (Figure 7). GO shows some individual graphene nano-sheets, but the most common are 4–10 stacked nano-sheets of less than 20 nm thick and 2–5 µm long. A-GO shows organized and highly stacked graphene nano-sheets whereas r-GO depicts large ultrathin and transparent silk curtain wave-like structure with a large number of wrinkles. Therefore, r-GO and, mainly, A-GO show larger number of stacked graphene nano-sheets than GO. This anticipates more net interactions between the polymers and GO than with r-GO and, particularly, A-GO.

In summary, the GO derivatives differ in surface chemistry and morphology. The GO surface contains the highest amount of oxygen functional groups, mainly C-O groups, and both r-GO and A-GO surfaces have lower and similar amounts of oxygen functional groups, but the amount of nitrogen is somewhat higher in A-GO. Furthermore, GO has a fewer number of stacked graphene nano-sheets, r-GO depicts large ultrathin wave-like stacked graphene nano-sheets, and A-GO shows the most organized and stacked graphene nano-sheets. Therefore, different extent of interactions of those GO derivatives with the waterborne poly(urethane urea) can be anticipated.

### 3.2. Characterization of the PUDs

The solids content and the pH values of the PUDs without and with 0.04 wt.% GO derivatives are given in Table 4. The solids content of the PUDs are 36–40 wt.%, they are quite close to the theoretical value. The pH value of the PUD containing A-GO is lower because of the existence of nitrogen surface groups on A-GO which causes the preferential creation of urea covalent bonds by reaction with the end isocyanate groups of the prepolymer during PUD synthesis, leading to lower ionic concentration on the particle surfaces. Thus, the differences in the pH values (8.5–9.8) of the PUDs can be ascribed to the functional groups on the GO derivative.

The variation of the viscosity at 25 °C as a function of the shear rate for the PUDs without and with 0.04 wt.% GO derivatives is shown in Figure 8. In general, at low shear rate, the viscosities of the PUDs differ significantly, but at high shear rate they are less different (19–34 mPa.s), this indicates the existence of non-Newtonian rheological behavior—shear thinning. The PUD without GO derivative shows the most noticeable shear thinning and the viscosity at a shear rate of 700 s^−1^ is 25 mPa.s. The addition of the GO derivative reduces the shear thinning in a different extent depending on its functional chemistry, only PUD+GO shows a Newtonian rheological behavior. Whereas PUD+A-GO shows some shear thinning and higher viscosity than PUD (34 mPa.s at a shear rate of 700 s^−1^), PU+r-GO exhibits more noticeable shear thinning but lower viscosity (19 mPa.s at a shear rate of 700 s^−1^). Because the extent of the shear thinning in the PUDs can be ascribed to the decrease of the ionic interactions between the particles, more net interactions between GO and the polyurethane chains are produced, likely due to the higher number of surface functional groups on this graphene oxide derivative.

### 3.3. Characterization of the PUs

The ATR-IR spectra of the poly(urethane urea)s (PUs) without and with 0.04 wt.% GO derivatives are given in Figure 9a. The typical absorption bands of the hard and the soft segments in the PUs can be distinguished in the ATR-IR spectra. The absorption bands due to the hard segments appear at 3364 cm^−1^ (N-H stretching), 1727 cm^−1^ (C=O stretching), and 1534 cm^−1^ (C-N stretching). The bands due to the soft segments correspond to C-H groups at 2855, 2873, and 2925 cm^−1^, CH_2_ group at 1461 cm^−1^, and several C-O bands of the polyester polyol at 1240, 1130, 1168, 1065, and 958 cm^−1^. The addition of 0.04 wt.% GO derivative does not produce new bands in the ATR-IR spectra of the PUs, but the ratio of the intensities of the C-O-C (soft segments) with respect to the one of the C=O (hard segments) stretching bands changes. Thus, according to Table 5, the I_C-O-C_/I_C=O_ ratio is higher in PU+A-GO and, to less extent, in PU+GO with respect to PU, indicating higher soft segments content in PU+A-GO and anticipating differences in the degree of micro-phase separation among the PUs containing different GO derivatives.

Figure 9b shows that the addition of GO or A-GO displaces the wavenumber of the N-H stretching band from 3364 cm^−1^ (PU without GO derivative) to 3351 or 3335 cm^−1^ respectively, revealing the interactions between the OH and/or NH_2_ functional groups on the GO and A-GO surfaces with the end NCO groups of the prepolymer that led to the formation of new urethane/urea hard segments. Furthermore, the intensity of the C-H stretching band at 2873 cm^−1^ decreases in PU+GO and PU+r-GO with respect to the other PUs, due to a change in the structure of the soft segments (Figure 9b). Similarly, the addition of GO and r-GO displaces the C-O band from 1240 cm^−1^ (PU without GO derivative) to 1257 cm^−1^ indicating again the structural change of the soft segments (Figure 9c). Therefore, the addition of the GO derivative with different surface chemistry affects differently the structure of the hard and soft segments, i.e., the degree of micro-phase separation, of the PUs.

Figure 10 shows the carbonyl region (1800–1600 cm^−1^) of the ATR-IR spectra of the PUs in which differences in the regions of the associated by hydrogen bond (H-bonded) urethane and urea groups are noticed. In order to quantify these differences, the curve fitting of the carbonyl region of the ATR-IR spectra of the PUs have been carried out. Appendix A shows, as typical example, the curve fitting of the carbonyl region of the PU without GO derivative in which four contributions due to the free urethane, H-bonded urethane, free urea, and H-bonded urea species can be distinguished.

Table 6 shows that the free urethane groups (1727–1726 cm^−1^) are dominant in all PUs and that the addition of the GO derivative decreases the percentage of the free urethane species and increases the percentage of the associated by hydrogen bond urethane species (1712–1708 cm^−1^), more noticeably in PU+r-GO. Furthermore, the amount of the free urea species (1693–1688 cm^−1^) is lower and the amount of the associated by hydrogen bond urea species (1677–1652 cm^−1^) is higher in PU+r-GO. Therefore, the structure of the hard segments changes by adding the GO derivative in a different manner depending on the surface functional groups, this should cause differences in the degree of micro-phase separation in the PUs.

Raman spectroscopy is more sensitive than IR spectroscopy due to the interactions between nanoparticles and polyurethane chains. Figure 11 shows some changes in the Raman shifts of the soft segments at 1124, 1090, and 1065 cm^−1^ (due to the C-N, C-O-C, and C-C groups respectively) indicating the existence of interactions between the soft segments of the poly(urethane urea) and the GO derivative.

The DSC traces corresponding to the second heating runs of the PUs are shown in Appendix A, and the thermal events in the DSC traces are given in Table 7. The PU without GO derivative shows the glass transition temperatures (T_g_s) of the soft segments at −53 °C and 5 °C, and the T_g_ of the hard segments at 193 °C. The addition of the GO derivative slightly increases the T_g_ of the soft segments and decreases the T_g_ of the hard segments (Table 7), this indicates lower degree of micro-phase separation which is somewhat more marked in PU+r-GO. Moreover, one melting peak at 45 °C of the soft segments appears only in PU+A-GO, because of its higher percentage of soft segments evidenced by ATR-IR spectroscopy—this indicates different interactions than in the other PUs. As a result, the addition of the GO derivative with different surface functionality changes differently the degree of the micro-phase separation in the PU.

The crystallinity of the PUs is due to the interactions between the hard and the soft segments, as assessed by X-ray diffraction. Figure 12 shows the X-ray diffractograms of the PUs in which three diffraction peaks at 2Ɵ values of 21.30–21.65, 22.20–22.35, and 23.75–24.10° can be distinguished, they are somewhat similar irrespective of the functional groups on the GO derivatives. The diffraction peaks at 2Ɵ values of 18.3–20.75° of the GO derivatives do not appear in the X-ray diffractograms of the PUs because of its adequate dispersion and low amount added. On the other hand, the intensities of the diffraction peaks are similar in all PUs, except in PU+GO in which more noticeable interactions between the hard and the soft segments are produced, likely caused by more net GO-poly(urethane urea) chains interactions.

The thermal stability and structure of the PUs were assessed by TGA. Figure 13a shows the variation of the weight as a function of the temperature of the PUs. PU+GO shows lower thermal stability than PU and the other PU+GO derivatives, likely due to more noticeable structural changes. The thermal stabilities of the PUs were quantified by the temperatures at which 5 (T_5%_) and 50 (T_50%_) wt.% are lost. According to Table 8, the addition of all GO derivatives decreases the value of T_5%_, due to the interactions between the hard and soft domains caused by the intercalation of the graphene nano-sheets. The lowest value of T_5%_ corresponds to PU+A-GO, as the soft segment content is higher than in the rest. On the other hand, the value of T_50%_ is similar in all PUs, except in PU+GO which is lower. Some literature has established the improved thermal stability of the polyurethane composites containing graphene derivatives [17,34,35], but the opposite trend is found in this study, this can be due to the small amount added of the GO derivatives and to the dominant effect of the GO derivative morphology of the nano-sheets in changing the structure of the hard and soft domains. Thus, the thermal stability of the PU is differently affected by adding GO derivatives with different functional groups.

The structural changes in the PUs caused by adding 0.04 wt.% GO derivatives can be better distinguished in the derivative TGA plots (Figure 13b). All PUs show one or two small thermal decompositions at 50–53 °C and 116–118 °C due to residual moisture (Table 9). The PU without GO derivative shows five additional thermal decompositions due to the urethane (280 °C) and urea (329 °C) hard domains, and the soft domains (372, 396, and 422 °C). All PUs containing GO derivative show an additional thermal decomposition at 190–206 °C with weight losses of 2–3 wt.%, this can be ascribed to the existence of GO derivative-poly(urethane urea) interactions. This thermal decomposition is produced at lower temperatures in PU+A-GO, this indicates less interactions between A-GO and the polymer chains. On the other hand, the temperature and the weight loss of the most thermal decompositions of the PU do not change by adding A-GO and r-GO, but the thermal decompositions of the soft domains at 372 and 422 °C do not appear as they are embedded in one unique thermal decomposition of the soft segments at 390–398 °C (Table 9). Still, the temperature of decomposition and the amount of the urea hard domains decrease in PU+GO due to stronger GO-polyurethane interactions (Table 9). This is in agreement with the different structure of PU+GO evidenced by ATR-IR spectroscopy, X-ray diffraction, and DSC.

The viscoelastic properties of the PUs have been assessed by DMA. Figure 14a indicates that all PUs show the glassy, glass transition, rubbery plateau, and melting regions. In the glassy region, the storage modulus of the PU increases by adding 0.04 wt.% GO derivatives, more noticeably in PU+GO and less markedly in PU+A-GO, which has higher percentage of soft segments. The glass transition and the rubbery plateau regions become steeper in the PUs with GO derivatives, except in PU+GO, and the melting is produced at a lower temperature. Therefore, the viscoelastic properties of the PUs are modified by adding GO derivatives and differently for PU+GO than for PU+A-GO and PU+r-GO. The viscoelastic behavior of PU+GO is somewhat similar to that of the PU without GO derivative, but the values of the storage moduli are higher due to the covalently bonded GO nano-sheets to the polymer chains. However, r-GO and, mainly, A-GO interact less effectively than GO with the polymer chains and disturb the interactions between the poly(urethane urea) chains leading to steeper glass transition and rubbery plateau regions.

Figure 14b shows the existence of two structural relaxations in the tan δ plots which can be associated to the glass transition temperatures of the PUs. The addition of the GO derivatives decreases the temperatures of the two structural relaxations (Table 10), this indicates better interactions between the soft and the hard segments, i.e., lower degree of phase separation, more noticeably in PU+r-GO and PU+A-GO. Furthermore, the value of the maximum of tan delta in PU+r-GO and PU+A-GO are significantly higher than in PU and PU+GO due to the intercalation of the GO derivative between the polymer chains that disturbs the interactions between the soft and the hard segments, and to the higher soft segments content in PU+A-GO. Therefore, the viscoelastic properties of the PUs are determined differently by adding the GO derivatives with different functional groups.

The dispersion of the GO derivatives particles in the poly(urethane urea) matrix in the PUs was assessed by confocal laser microscopy (Figure 15). All GO derivatives are well dispersed into the PU matrix, the r-GO and A-GO particles have similar size (4–8 µm) and they do not agglomerate. However, most GO particles cannot be distinguished in PU+GO but some agglomerates of nano-sheets can be noticed.

Because of the different polarity of the polyurethane and the GO derivatives with different surface chemistry, the wettability of the PU+GO derivative surface may change. The water contact angle on the PU without GO derivative surface is 47 degrees, and the addition of r-GO or GO does not change the wettability (47–48 degrees). Conversely, the addition of A-GO decreases the water contact angle value to 33 degrees, likely due to the high polarity of the nitrogen functional groups, this is in agreement with the lower pH obtained in PUD+A-GO.

The addition of the GO derivatives also affects the mechanical properties of the PUs which were assessed by stress-strain tests. According to Figure 16, all stress-strain curves show a noticeable yield point (less marked in PU+A-GO) followed by the plastic deformation until the PU breaks. The addition of the GO derivatives increases the elongation-at-break (Table 11) and decreases the tensile strength and the yield stress of the PU. PU+GO shows higher elongation-at-break and tensile strength values than PU+r-GO and PU+A-GO. This trend agrees with the better viscoelastic properties and degree of micro-phase separation of PU+GO with respect to PU+r-GO and PU+A-GO. Furthermore, PU+r-GO shows similar yield stress to PU (Table 11) because of larger and thicker nano-sheets of r-GO which may cause fewer net interactions with the polymer chains. Therefore, the addition of the GO derivatives seems to impart some toughening to the PU, to a different extent depending on the size of the nano-sheets and the functional groups of the GO derivatives.

The addition of the carbon nanoparticles strongly influences the morphologies of the waterborne polyurethanes as has been observed for GO-PU [36], TiO_2_/r-GO-PU [37], and PU-graphene [38] composites. Therefore, the morphology of the PU should change when the GO derivatives are added—this was analyzed by SEM micrographs of the fractured surfaces of the PUs. Whereas PU without GO exhibits relativity smooth fractured surface as expected for an elastomeric material (Figure 17), all PUs containing the GO derivatives show rough fractured surfaces indicating some toughness (as anticipated by the increased elongation-at-break values obtained in the stress-strain tests). The appearance of the fractured surfaces differs depending on the GO derivative. While a rough micro-fractured surface and some small cracks and voids are seen in PU+r-GO, a rough wrinkled and bumped fractured surface with some A-GO nano-sheets and a few cracks are distinguished in PU+A-GO. The fractured surface of PU+GO is less rough, no cracks but small fractured zones are distinguished and the adhesion of the GO nano-sheets to the fractured PU matrix is good. Therefore, the addition of the different GO derivatives causes different morphologies of the PUs. Similar findings have been reported in our recent work dealing with PUs containing different graphene derivatives [39].

The functional groups (C-O, N-H) on the surface and edges of the GO derivatives covalently attach to the prepolymer chains during PUD synthesis. The interactions between the poly(urethane urea) chains with covalently bonded GO derivative nano-sheets changes the degree of micro-phase separation of the PU: these changes produce different thermal, viscoelastic, mechanical, and surface properties, i.e., different structure–property relationship are produced. During PUD synthesis, the oxygen functional groups on the GO and r-GO surface produce new urethane linkages by the reaction with the end NCO groups of the prepolymer chain (Figure 18), to a greater extent in PU+GO than in PU+r-GO because of the higher amount of oxygen functional groups on the GO surface. Furthermore, the number of stacked graphene nano-sheets in r-GO is higher than in GO. Because the nano-sheets intercalate among the polymer chains, the degree of micro-phase separation will be different in PU+GO and PU+r-GO. GO has the highest number of functional groups and the thinnest graphene nano-sheets, so more net interactions with the polymer chains will be produced. Conversely, r-GO has the lowest content of functional groups and thicker stacked graphene nano-sheets than GO, so the improvement of the properties in PU+r-GO should be less pronounced.

A-GO surface had a similar amount of oxygen functional groups to r-GO and also some nitrogen functional groups. Because of the higher reactivity of the amines with the isocyanates, during PUD synthesis, the amine functional groups on the A-GO surface should react preferentially, producing new urea hard domains (Figure 18); additionally, there is a chance that some oxygen functional groups on the A-GO surface may also react with the NCO groups. Furthermore, the number of stacked graphene nano-sheets is higher in A-GO than in GO, so the intercalation of the covalently A-GO nano-sheets will disturb differently the interactions between the poly(urethane urea) chains. As a consequence, the degree of micro-phase separation of PU+A-GO differs from the ones in PU+GO and PU+r-GO, this justifies the reduced increase in properties, i.e., PU+A-GO has the lowest thermal stability, the lowest temperature of the thermal decomposition of the A-GO/PU interactions, and the highest value of the maximum of tan delta, the highest surface polarity, and the less improved mechanical properties.

### 3.4. Adhesion Properties of the PUDs

The adhesion of the PUDs containing the GO derivatives with different functional groups has not been addressed yet in the existing literature. In this study, the adhesion properties were determined by T-peel tests. Table 12 shows the values of the T-peel strength of the plasticized PVC/PUD/ plasticized PVC joints determined 1 and 72 h after joint formation. One hour after joint formation, i.e., immediate adhesion, the T-peel strength values are significantly lower in the joints made with the PUDs containing A-GO and r-GO, but the T-peel strength is higher in the one made with PUD+GO. Because the water in the adhesive is not completely removed after one hour of the joint formation, a cohesive failure of the adhesive is obtained in all joints. However, 72 h after joint formation, i.e., final adhesion, an increase in the T-peel strength is obtained in all joints due to the complete water removal. The highest T-peel strength values correspond to the joints made with PUD+GO and PUD+r-GO, and a rupture of the PVC substrate is obtained, indicating an optimal adhesion; on the other hand, similar lower T-peel strength values are obtained in the joints made with PUD and PUD+A-GO.

The improvement of the adhesion in the joints made with PUD+GO and PUD+r-GO can be related to the more net covalent GO derivative/PU interactions. Due to the existence of thinner graphene nano-sheets in GO, the immediate adhesion is higher in the joints made with PUD+GO; however, the final adhesion is similar in the joints made with PUD+GO and PUD+r-GO, this indicates that the final adhesion is mainly determined by the covalent linkages between the oxygen functional groups in the GO derivative surface and the prepolymer chains. Conversely, the lower adhesion of the joints made with PUD+A-GO is consistent with its higher content of the soft segments and the lower covalent linkages between the oxygen functional groups on the GO derivative surface and the prepolymer chains.

## 4. Conclusions

GO derivatives with different surface chemistry and morphology were successfully added for in-situ polymerization before prepolymer formation during the synthesis of waterborne poly(urethane urea)s. The GO surface had the highest amount of oxygen functional groups, mainly C-O groups, and both r-GO and A-GO surfaces had lower and similar amounts of oxygen functional groups, A-GO also contained nitrogen functional groups. Furthermore, GO was composed of 4–10 stacked graphene nano-sheets, r-GO depicted large ultrathin wave-like stacked graphene nano-sheets, and A-GO showed the most organized and stacked graphene nano-sheets.

During PUD synthesis, new urethane groups were formed by reacting the C-O groups on the GO and r-GO surface with the end NCO groups of the prepolymer, the GO and r-GO nano-sheets were covalently bonded to the urethane hard segments. Consequently, lower degrees of phase separation in PU+GO and in PU+r-GO than in PU were obtained, leading to increased percentage of associated by hydrogen bond urethane species and stronger interactions between the soft segments. Because of the lower amount of oxygen functional groups and thicker stacked graphene nano-sheets in r-GO, the degree of phase separation was lower in PU+r-GO than in PU+GO. On the other hand, the amine functional groups on the A-GO surface will react preferentially with the end NCO groups of the prepolymer during PUD synthesis producing new urea hard domains. Furthermore, because A-GO had thicker stacked graphene nano-sheets, a higher percentage of soft segments than in PU without GO derivative was produced, which led to less net A-GO/PU interactions.

The pH value of the PUD containing A-GO was lower than in the other PUDs due to lower ionic concentration on the particle surfaces, and thus, the differences in the pH values (8.5–9.8) can be ascribed to the functional groups on the GO derivative. The PUD without GO derivative showed the most noticeable shear thinning and the addition of the GO derivatives reduced differently the extent of the shear thinning depending on their functional chemistry, PUD+GO showed a Newtonian rheological behavior. Because the extent of the shear thinning in the PUDs was ascribed to decreased ionic interactions between the particles, more net interactions between GO with the poly(urethane urea) chains were produced.

All GO derivatives were well dispersed into the PU matrix. The addition of A-GO increased more the percentage of the soft segments in PU+A-GO than in the other PUs. In fact, PU+A-GO was the only one to show a melting peak of the soft segments in the DSC traces. Furthermore, the structure and interactions of the soft segments changed in PU+GO and PU+r-GO (lower intensity of the C-H stretching band at 2873 cm^−1^, C-O band displaced to higher wavenumber). The free urethane species were dominant in the hard segments of all PUs and the addition of the GO derivatives increased the percentage of the associated by hydrogen bond urethane species, more noticeably in PU+r-GO. Consequently, the addition of the GO derivatives slightly increased the T_g_ of the soft segments and decreased the T_g_ of the hard segments of the PU, indicating a lower degree of micro-phase separation which was somewhat more marked in PU+r-GO.

The thermal stability of the PU was differently affected by adding the GO derivatives with different surface functional groups. PU+GO showed lower thermal stability than the other PUs, likely due to more noticeable structural changes. The addition of all GO derivatives decreased the value of T_5%_, due to the interactions between the hard and soft domains caused by the intercalation of the graphene nano-sheets, the lowest value of T_5%_ corresponds to PU+A-GO, with a soft segment content higher than in the rest. All PUs containing the GO derivatives exhibited an additional thermal decomposition at 190–206 °C which was ascribed to the GO derivative-poly(urethane urea) interactions, the lowest temperature corresponded to PU+A-GO in which lower A-GO/polymer interactions were produced.

The viscoelastic properties of the PUs were also determined differently by adding the GO derivatives with different surface functional groups. In the glassy region, the storage modulus of the PU increased and the melting was produced at a lower temperature by adding the GO derivatives, more noticeably in PU+GO and less markedly in PU+A-GO due to its higher percentage of soft segments. However, the PUs exhibited two structural relaxations which temperatures decreased by adding the GO derivatives, more noticeably in PU+r-GO and PU+A-GO. Furthermore, the value of the maximum of tan delta in PU+r-GO and PU+A-GO are significantly higher than in PU and PU+GO, more noticeably in PU+A-GO in which the higher soft segments content contributed to higher viscous modulus.

The addition of the GO derivatives also affected the mechanical properties of the PUs, as the elongation-at-break increased although the tensile strength and the yield stress decreased. PU+GO showed higher elongation-at-break and tensile strength values than PU+r-GO and PU+A-GO. The addition of the GO derivatives imparted some toughness to the PU, to a different extent depending on the number of surface functional groups and stacked graphene nano-sheets in the GO derivative. The toughening of the PUs containing the GO derivatives was confirmed by the roughened fractured surfaces found in the SEM micrographs.

The addition of the GO derivatives also affected the surface and adhesion properties of the PUs. The water contact angle on the PU did not change by adding r-GO or GO, but a decrease was obtained in PU+A-GO likely due to the high polarity of the nitrogen functional groups. The highest T-peel strength values corresponded to the joints made with PUD+GO and PUD+r-GO, and a rupture of the PVC substrate was obtained, indicating an optimal adhesion; on the other hand, similar lower T-peel strength values were obtained in the joints made with PUD and PUD+A-GO.

Additional future studies are planned on the use of different PUD formulations and the addition of different graphene-based materials and inorganic oxide blends for improving their thermal stability without affecting their adhesion. Furthermore, the electrical conductivities of the PUDs containing different graphene derivatives and their corrosion resistance in NaCl solutions of PUD coatings on different metals are under study.

## Figures and Tables

**Figure 1 materials-14-04377-f001:**
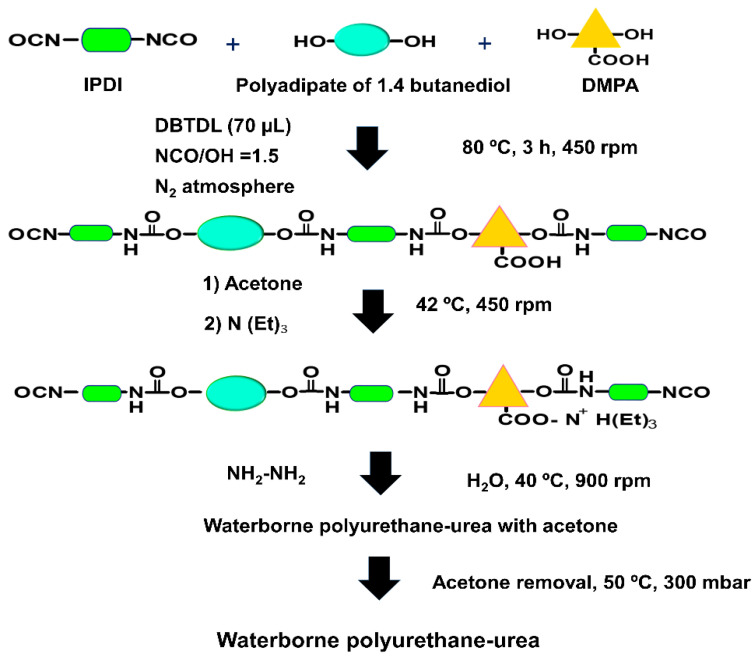
Scheme of the synthesis of the waterborne poly(urethane urea) dispersion.

**Figure 2 materials-14-04377-f002:**
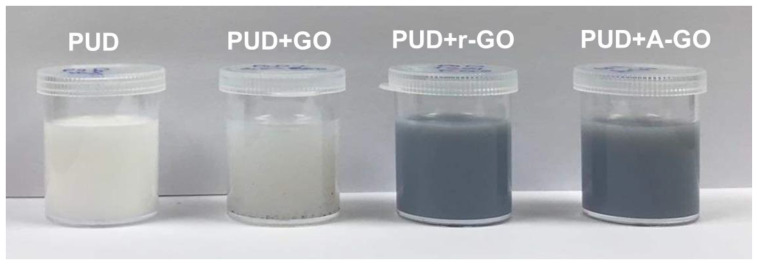
Appearance of the PUDs without and with 0.04 wt.% GO derivative, one month after synthesis.

**Figure 3 materials-14-04377-f003:**
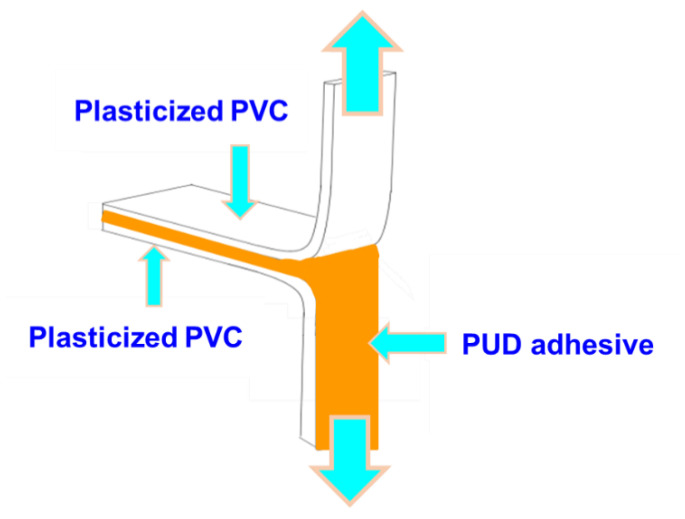
Scheme of the T-peel test of plasticized PVC/PUD adhesive/plasticized PVC joints.

**Figure 4 materials-14-04377-f004:**
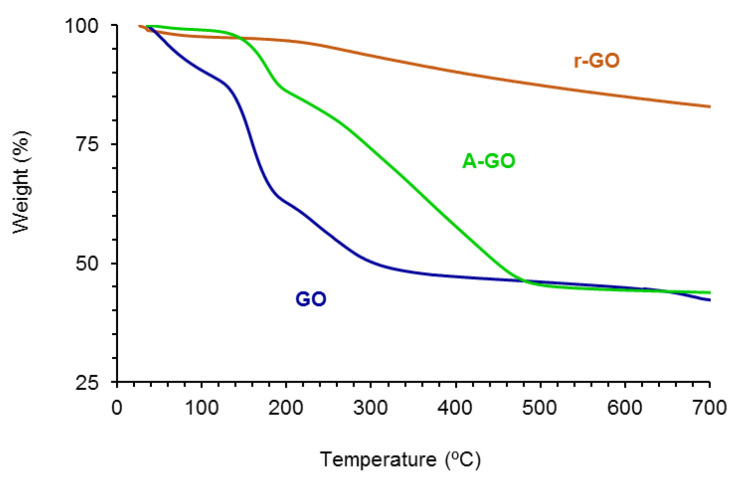
Variation of the weight of the graphene oxide derivatives as a function of the temperature. TGA experiment.

**Figure 5 materials-14-04377-f005:**
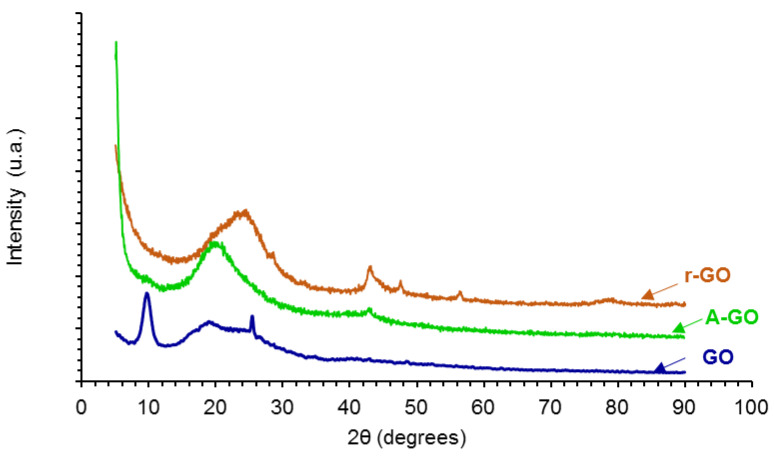
X-ray diffractograms of the graphene oxide derivatives.

**Figure 6 materials-14-04377-f006:**
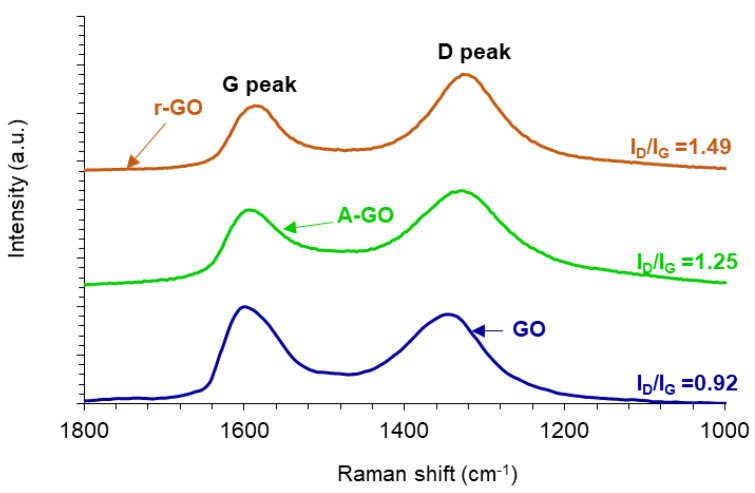
Raman spectra of the graphene oxide derivatives.

**Figure 7 materials-14-04377-f007:**
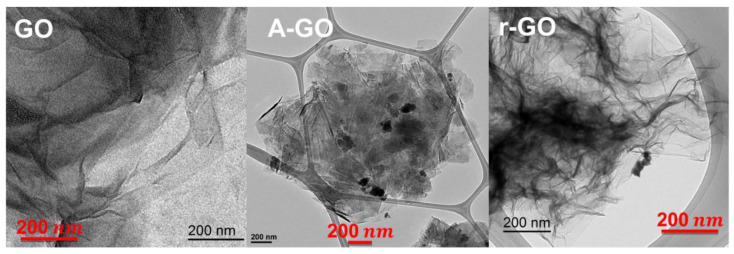
TEM micrographs of the graphene oxide derivatives on carbon grid.

**Figure 8 materials-14-04377-f008:**
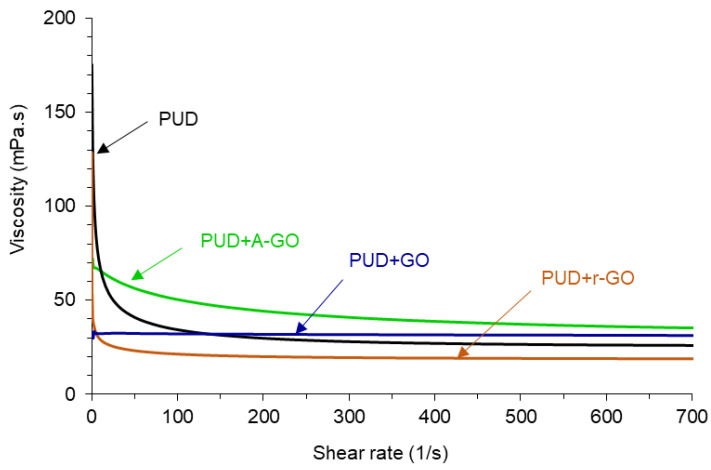
Variation of the viscosity at 25 °C of the PUDs as a function of the shear rate.

**Figure 9 materials-14-04377-f009:**
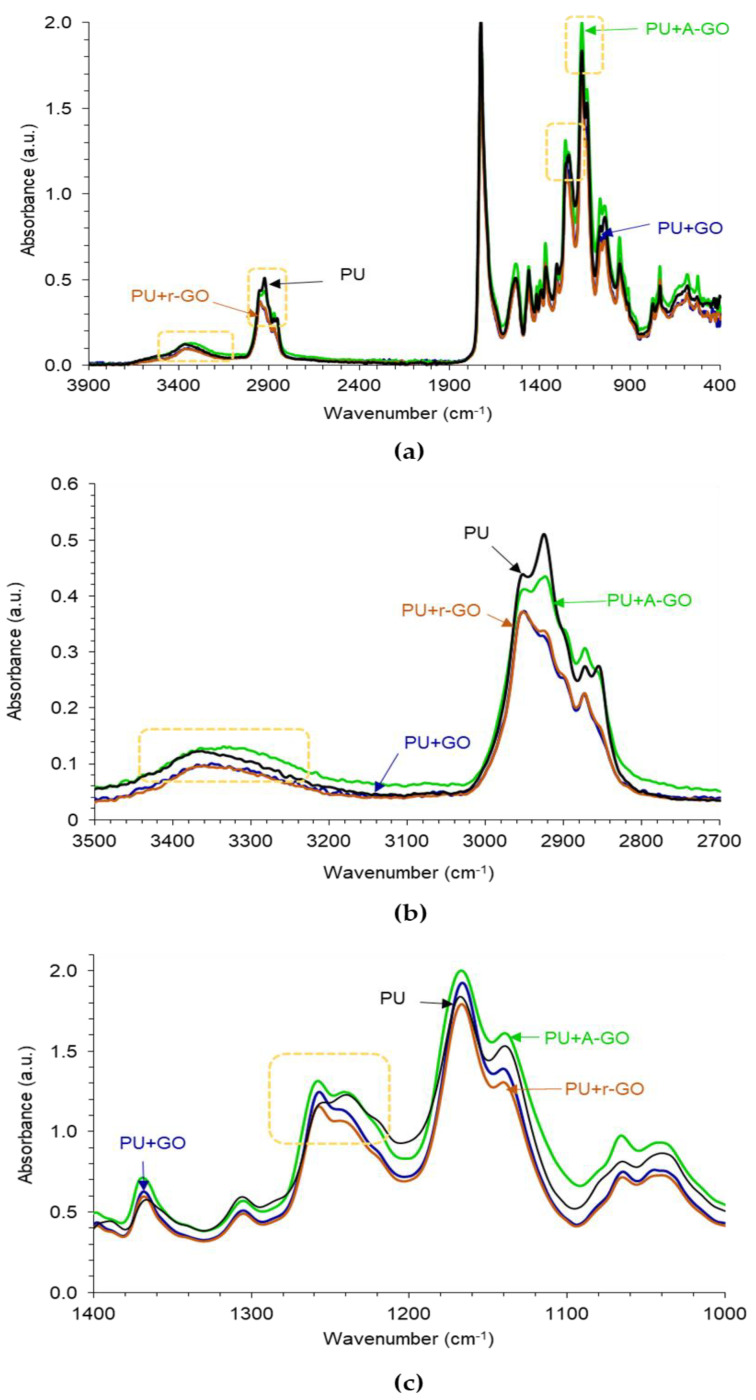
(**a**) ATR-IR spectra of the PUs without and with GO derivatives; (**b**) 3500–2700 cm^−1^ region, and (**c**). 1400–1000 cm^−1^ region of the ATR-IR spectra.

**Figure 10 materials-14-04377-f010:**
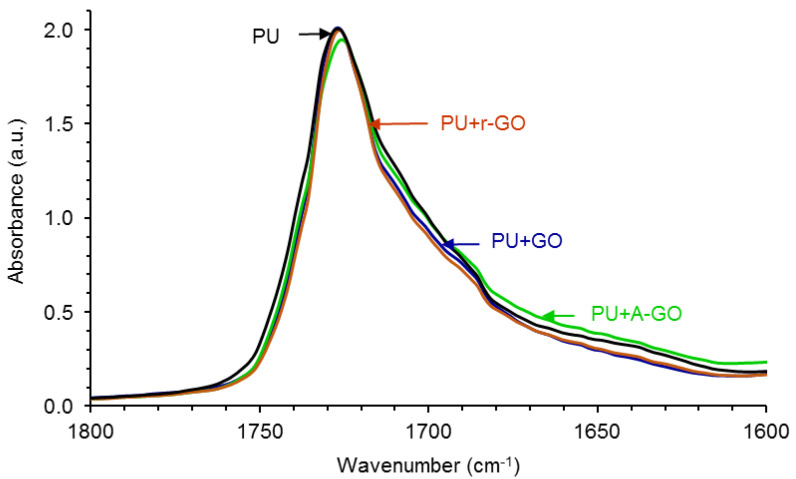
1800–1600 cm^−1^ region of the ATR-IR spectra of the PUs without and with GO derivatives.

**Figure 11 materials-14-04377-f011:**
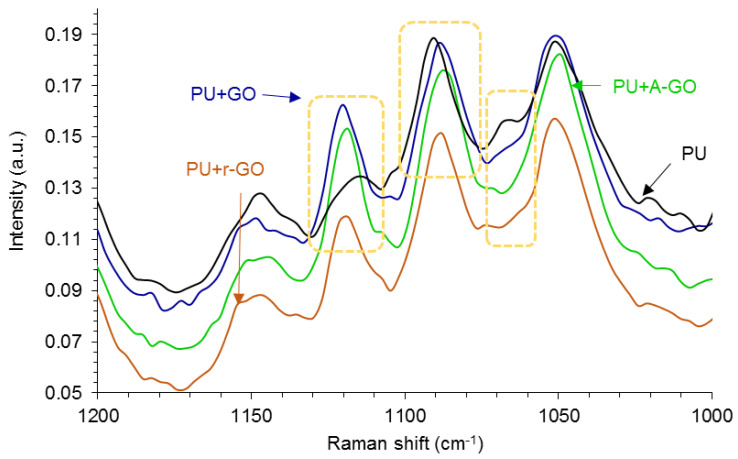
1200–1000 cm^−1^ region of the Raman spectra of the PUs without and with GO derivatives.

**Figure 12 materials-14-04377-f012:**
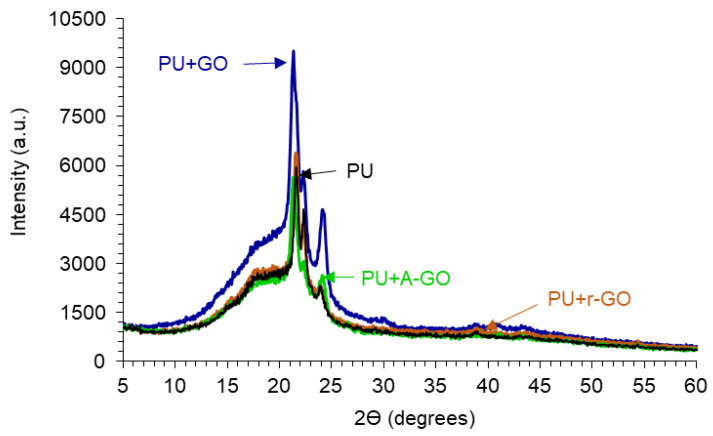
X-ray diffractograms of the PUs without and with GO derivatives.

**Figure 13 materials-14-04377-f013:**
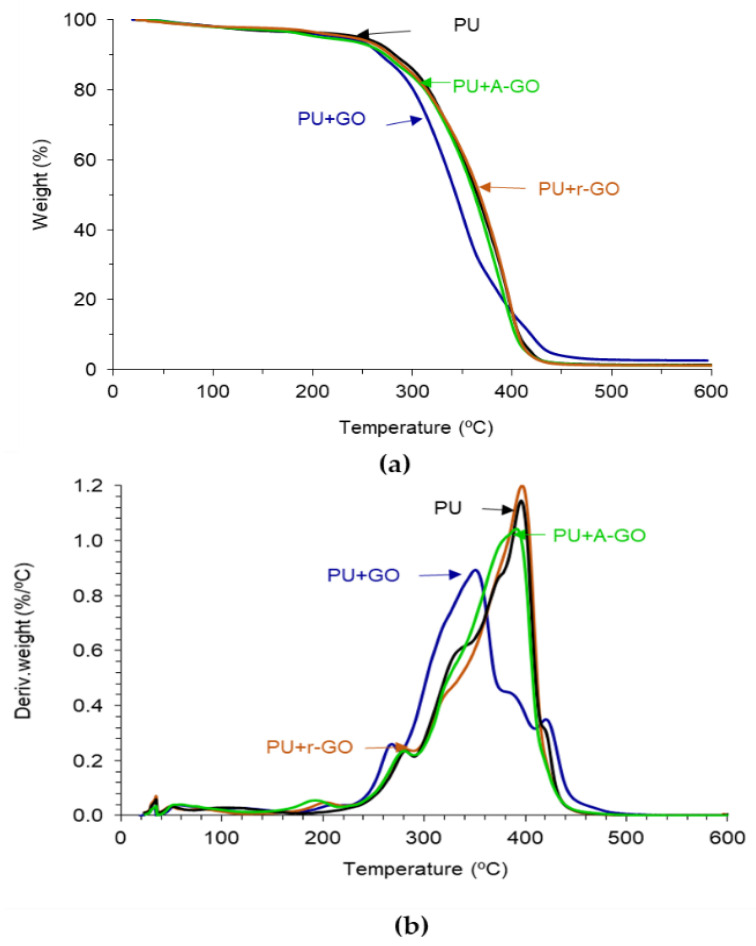
Variation of (**a**) the weight and (**b**) the derivative of the weight of the PUs as a function of the temperature. TGA experiment.

**Figure 14 materials-14-04377-f014:**
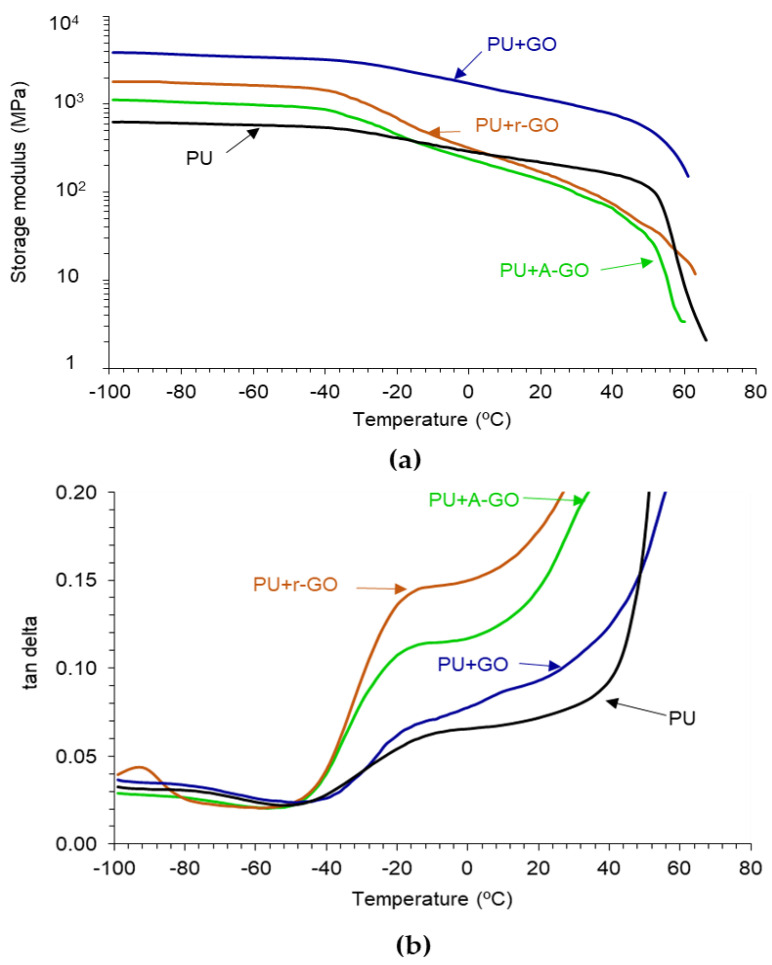
(**a**) Variation of the storage modulus (E′) as a function of the temperature for PUs; (**b**) Variation of tan delta as a function of the temperature for PUs. DMA experiments.

**Figure 15 materials-14-04377-f015:**
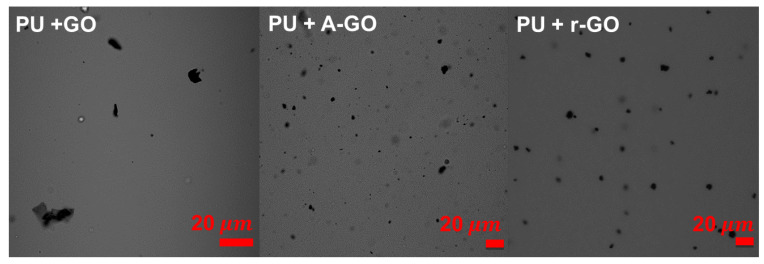
Confocal laser micrographs of the PUs with GO derivatives.

**Figure 16 materials-14-04377-f016:**
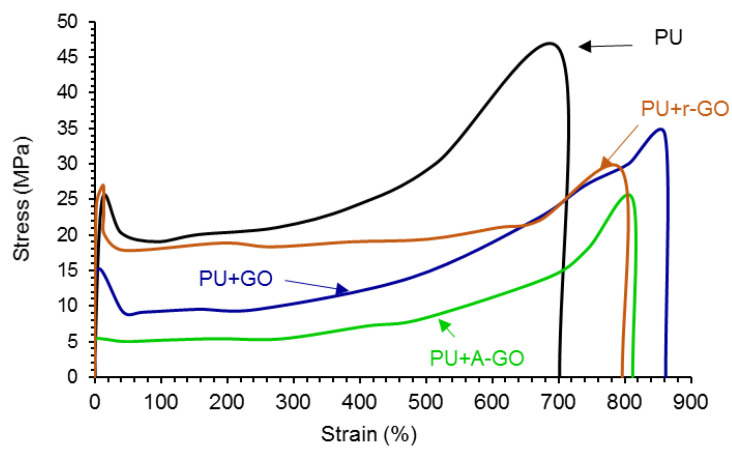
Stress-strain curves of the PUs.

**Figure 17 materials-14-04377-f017:**
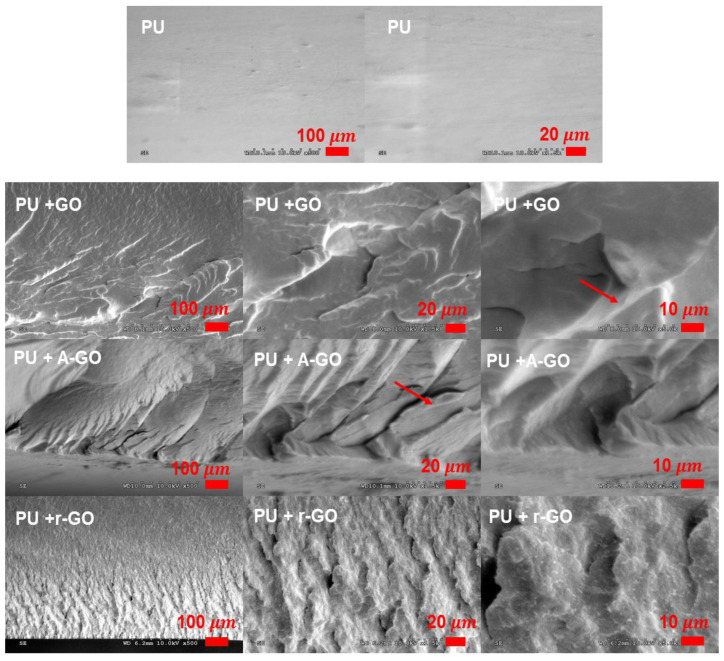
SEM micrographs of the fractured surfaces of the PUs. Red arrows show the location of the GO derivative sheets.

**Figure 18 materials-14-04377-f018:**
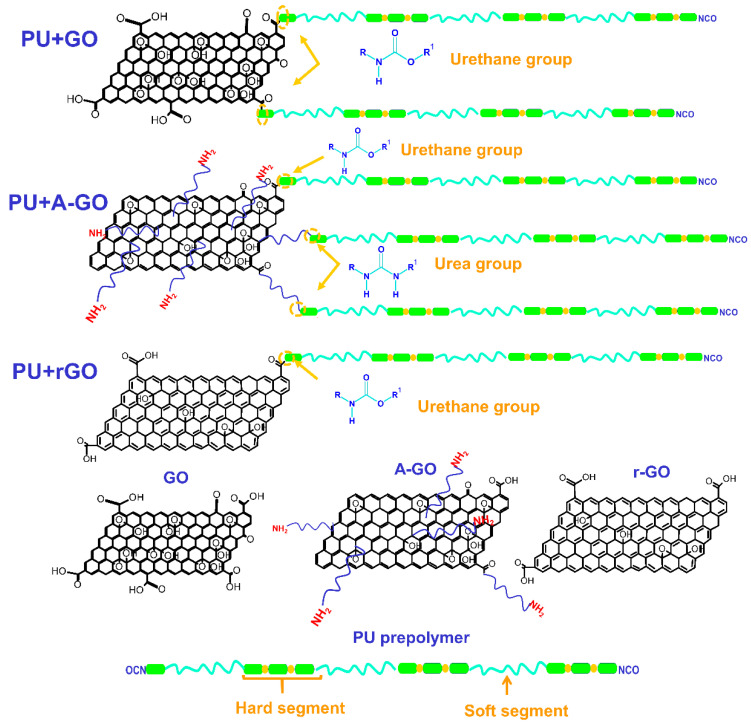
Scheme of the covalent linkages between the functional groups on the GO derivatives surfaces and the end NCO groups of the prepolymer produced during PUD synthesis.

**Table 1 materials-14-04377-t001:** Atomic percentages of elements on the graphene oxide derivative surfaces. XPS experiments.

GO Derivative	Carbon (at.%)	Oxygen (at.%)	Nitrogen (at.%)
GO	64 ± 0.5	35 ± 0.5	<1
A-GO	84 ± 0.5	14 ± 0.5	2
r-GO	87 ± 0.5	12 ± 0.5	<1

**Table 2 materials-14-04377-t002:** Binding energies and percentages of species on the graphene oxide derivative surfaces. C1s spectra.

GO Derivative	Species	Percentage (at.%)	Binding Energy (eV)
GO	C-C	62	284.6
	C-O	31	286.8
	C=O	5	288.5
	O-C=O	2	290.1
A-GO	C-C	72	284.6
	C-O, C-N	23	286.2
	C=O	5	288.7
r-GO	C-C	67	284.6
	C-O	24	285.9
	C=O	8	288.4
	O-C=O	1	290.7

**Table 3 materials-14-04377-t003:** Amounts of functional groups of the graphene oxide derivatives. TGA experiment.

GO Derivative	Group	T (°C)	Weight Loss (%)
GO	Moisture	59	14
	-C-OH	158, 231–257	41
	-O-C=O	676–688	6
A-GO	Moisture	50	1
	-C-N	177	18
	-C-OH	263	11
	C=O	361, 438	26
r-GO	Moisture	54	3
	-C-OH	258	15

**Table 4 materials-14-04377-t004:** Solids content and pH values of the PUDs.

Poly(urethane urea) Dispersion	Solids Content (wt.%) ^a^	pH
PUD	38.5 ± 0.5	9.8 ± 0.0
PUD+GO	36.1 ± 0.3	9.6 ± 0.0
PUD+A-GO	39.0 ± 1.1	8.5 ± 0.0
PUD+r-GO	39.6 ± 0.8	9.1 ± 0.0

^a^ Theoretical solids content: 40 wt.%.

**Table 5 materials-14-04377-t005:** Values of the ratios of the intensities of the C-O-C bending (1168 cm^−1^) with respect to the one of the C=O stretching (1727 cm^−1^) bands in the ATR-IR spectra of the PUs.

Poly(urethane urea)	I_C-O-C_/I_C=O_
PU	0.91
PU+GO	0.94
PU+A-GO	1.04
PU+r-GO	0.89

**Table 6 materials-14-04377-t006:** Relative contributions of the free and associated by hydrogen bond (H-bonded) urethane and urea species in the PUs. Curve fitting of the carbonyl region of the ATR-IR spectra.

Wavenumber (cm^−1^)	Relative Contribution of Species (%)			
	PU	PU+GO	PU+A-GO	PU+r-GO
1727–1726 (free urethane)	54	43	43	40
1712–1708 (H-bonded urethane)	19	30	33	39
1693–1688 (free urea)	17	19	16	9
1657–1652 (H-bonded urea)	10	8	8	12

**Table 7 materials-14-04377-t007:** Thermal events obtained from the DSC traces of the PUs. Second heating run.

Poly(urethane urea)	T_g1_ (°C)	T_g2_ (°C)	T_g3_ (°C)	T_m_ (°C)	∆H_m_ (J/g)
PU	−53	5	193	-	-
PU+GO	−50	5	178	-	-
PU+A-GO	−52	5	179	45	1
PU+r-GO	−51	4	167	-	-

**Table 8 materials-14-04377-t008:** Values of the temperatures at which 5 (T_5%_) and 50 (T_50%_) wt.% are lost in the PUs. TGA experiments.

Poly(urethane urea)	T_5%_ (°C)	T_50%_ (°C)
PU	247	365
PU+GO	227	343
PU+A-GO	206	362
PU+r-GO	237	367

**Table 9 materials-14-04377-t009:** Values of the temperatures and the weight losses of the thermal decompositions of the PUs. At the end of TGA experiment, all PUs exhibit a residue of 1 wt.%.

Poly(urethane urea)	PU	PU+GO	PU+A-GO	PU+r-GO
T_1_ (°C)	50,118	56,116	53	50
Weight loss_1_ (%)	3	4	3	2
T_2_ (°C)	-	206	190	202
Weight loss_2_ (%)	-	2	2	2
T_3_ (°C)	280	266	278	278
Weight loss_3_ (%)	8	7	10	10
T_4_ (°C)	329	315	320	318
Weight loss_4_ (%)	29	25	19	20
T_5_ (°C)	372	350	-	-
Weight loss_5_ (%)	23	36	-	-
T_6_ (°C)	39	389	390	398
Weight loss_6_ (%)	31	15	64	64
T_7_ (°C)	422	422	-	-
Weight loss_7_ (%)	5	10	-	-

**Table 10 materials-14-04377-t010:** Values of the glass transition temperature and the maximum of tan delta of the PUs. DMA experiments.

Poly(urethane urea)	T_1_ (°C)	Max tan δ_1_	T_2_ (°C)	Max tan δ_2_
PU	−76	0.03	−6	0.08
PU+GO	−87	0.03	−11	0.07
PU+A-GO	−80	0.03	−16	0.12
PU+r-GO	−90	0.04	−15	0.15

**Table 11 materials-14-04377-t011:** Some mechanical properties obtained from the stress-strain curves of the PUs.

Poly(urethane urea)	Yield Stress (MPa)	Tensile Strength (MPa)	Elongation-at-Break (%)
PU	27 ± 6	48 ± 21	696 ± 100
PU+GO	13 ± 4	34 ± 9	846 ± 9
PU+A-GO	5 ± 0	25 ± 8	814 ± 30
PU+r-GO	28 ± 8	29 ± 10	796 ± 79

**Table 12 materials-14-04377-t012:** T-peel strength values of the plasticized PVC/PUD without and with GO derivative/plasticized PVC joints obtained 1 and 72 h after joint formation. Locus of failure: CS—Cohesive failure of the substrate surface; CA—Cohesive failure of the adhesive; S—Rupture of the PVC substrate.

Poly(urethane urea) Dispersion	T-Peel Strength—1 h (kN/m)	Locus of Failure—1 h	T-Peel Strength—72 h (kN/m)	Locus of Failure—72 h
PUD	5.4 ± 1.2	CA	7.0 ± 1.0	CA+CS
PUD+GO	6.4 ± 0.7	CA	13.5 ± 0.7	S
PUD+A-GO	2.6 ± 0.5	CA	7.6 ± 2.1	CS+CA+S
PUD+r-GO	3.0 ± 0.3	CA	12.6 ± 2.0	CS

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
