# Peer review of "Influence of the Surface Chemistry of Graphene Oxide on the Structure–Property Relationship of Waterborne Poly(urethane urea) Adhesive"

_materials, 2021, doi:10.3390/ma14164377_

Round 1

Reviewer 1 Report

The manuscript describes the influence of the surface chemistry of graphene oxide on the structure-property relationship of waterborne poly(urethane urea) adhesive. The manuscript presented concerns an interesting and actual subject. This manuscript can be accepted after major revision.

Some comments are listed below:

  1. Please seek guidance from a native English speaker if possible. The overall English needs to be improved: words, commas, "the" etc.
  2. The introduction is well-written but not sufficient. More papers need to be cited and discussed. Please add some information about the presence of functional groups which is important for medical or electrochemical applications and others applications. Please cite: (1) Nanomaterials. 2019; 9 (12): 1758. DOI: 10.3390/nano9121758 (2) Materials 2020, 13(21), 4975; https://doi.org/10.3390/ma13214975 (3) Chemical Engineering Journal, 389, 2020, 124375, https://doi.org/10.1016/j.cej.2020.124375
  3. Please correct Figure 18 to better quality (especially structural formulas).
  4. Please correct the scale in Figure 15 and Figure 17 (red stripe, we can see white inscriptions too).
  5. Please add a figure with high-resolution X-ray photoelectron spectra for C1s, N1s, and O1s of representative samples.
  6. Could the authors include the standard deviation of the statistical analysis in XPS?
  7. Please explain the low concentration of the C element for GO in comparison to other samples.
  8. Why authors use commercial substrates. So, what is the novelty in synthesizing GO by San Sebastian groups (Graphenea) and Synthesia (Barcelona)? Similarly, what is the novelty in preparing rGO? Please explain that in the comments.
  9. Please add more sentences to the discussion of the characterization by SEM. Is EDS a good method to analyze elements such as C or O? Please explain.
  10. Authors are suggested to describe some future plans in conclusions.

Reviewer 2 Report

This is a comprehensive study of the influence of surface functional groups of GO on the property of PUDs. Authors provided a lists of characterizations of PUDs prepared by adding GO, rGO and AGO, revealing the relationship between the surface chemistry of GO and GO+PUDs. I recommend the publication of this paper in this journal after minor revision.

1). In FigS1 and table 4. the decomposition occurred at 158 C is attributed to C-OH. Why this decomposition process is not observed in AGO and rGO, since they also have C-OH function group.

2).In Fig5. Why rGo and AGo have different scan range? especially there is a peak observed at 9.55 in GO. The trends of r-go and a-go may also have a peak at the same position. 

3). Also in Fig5. Why ago and rgo have peaks at 42.9,47.55,56,55 and 77.1 degree? What makes these peaks absent in GO xrd pattern?  what about the peak at ~25 in GO? what is  that peak attribute to and why absent in the other two samples?

4) the PH of PUD+ago is 8.5. This is a huge difference from others. How come 0.04 wt% AGO makes such a huge differencec?

5) Why the content of GO is 0.04%. How things change when increasing the GO amount? 

Author Response

Please see the atatched file

Round 2

Reviewer 1 Report

Accept in the present form